# Ethnic differences in respiratory disease for Native Hawaiians and Pacific Islanders: Analysis of mediation processes in two community samples

**Thomas A. Wills**[1]*, **Joseph Keawe'aimoku Kaholokula**[2], **Pallav Pokhrel**[1], **Ian Pagano**[1]

**1** Cancer Prevention in the Pacific Program, University of Hawaii Cancer Center, Honolulu, Hawaii, United States of America, **2** Department of Native Hawaiian Health, John A. Burns School of Medicine, University of Hawaii at Manoa, Honolulu, Hawaii, United States of America

* twills@cc.hawaii.edu

**Data Availability Statement:** The data are available by request to the Hawaii State Department of Health, Hawaii Health Data Warehouse: https://hhdw.org/. The data are owned by a third party, the

## Abstract

### Objective

The prevalence of asthma and chronic obstructive pulmonary disorder (COPD) is elevated for Native Hawaiians but the basis for this differential is not well understood. We analyze data on asthma and COPD in two samples including Native Hawaiians Pacific Islanders, and Filipinos to determine how ethnicity is related to respiratory disease outcomes.

### Methods

We analyzed the 2016 and 2018 Behavioral Risk Factor Surveillance Survey (BRFSS), a telephone survey of participants ages 18 and over in the State of Hawaii. Criterion variables were a diagnosis of asthma or COPD by a health professional. Structural equation modeling tested how five hypothesized risk factors (cigarette smoking, e-cigarette use, second-hand smoke exposure, obesity, and financial stress) mediated the ethnic differential in the likelihood of disease. Age, sex, and education were included as covariates.

### Results

Structural modeling with 2016 data showed that Native Hawaiian ethnicity was related to higher levels of the five risk factors and each risk factor was related to a higher likelihood of respiratory disease. Indirect effects were statistically significant in almost all cases, with direct effects to asthma and COPD also observed. Mediation effects through comparable pathways were also noted for Pacific Islanders and Filipinos. These findings were replicated with data from the 2018 survey.

### Conclusions

Native Hawaiian and Pacific Islander ethnicity is associated with greater exposure to five risk factors and this accounts in part for the ethnic differential in respiratory disease

Hawaii State Department of Health. There are some restrictions on the data provided: HHDW will not release data on zip codes or certain other demographic information that could be used to identify an individual participant. Zip codes are restricted because Hawaii is a relatively small state with diverse ethnicity, and a person who had data on ethnicity and zip code could possibly identify a participant through deductive identification (e.g., the only female Tongan in the Kona Zip code). If there are any questions about this they can be directed to Dr Tonya Lowery St. John (tlowerystjohn@hhdw.org).

**Funding:** This work was supported in part by grant #P30 CA071789 from the National Cancer Institute (TAW), grant #U54 GM138062 from the National Institute of General Medical Sciences (KK), and by grants # R01 CA228905 from the National Cancer Institute and #R01 DA053766 from the National Institute on Drug Abuse (PP). The funding agencies played no role in analysis and interpretation of the data, preparation of the manuscript, and decision to submit the manuscript for publication. The content is solely the responsibility of the authors and does not necessarily reflect official policy of the U.S. Department of Health and Human Services.

**Competing interests:** The authors declared that no competing interests exist.

outcomes. The results support a social-ecological model of health disparities in this population. Implications of the findings for preventive interventions are discussed.

## Introduction

This research examines asthma and chronic obstructive pulmonary disease (COPD) in Native Hawaiians and Pacific Islanders. Recent commentaries have observed that there is a marked underrepresentation of these populations in health research and have noted that this can undermine health equity [1,2]. Addressing this issue requires disaggregating groups that have previously been combined in prevalence research [3] and testing for differential exposures that may produce disparities in health or disease in these populations [1]. We address this by reporting disaggregated data on the prevalence of respiratory disease for Native Hawaiians and other racial/ethnic groups in the population of Hawaii, delineating ethnic differences in exposure to risk factors for these conditions, and analyzing how ethnic differences in exposure are linked to respiratory outcomes.

Existing data have indicated that Native Hawaiians have the highest mortality rate for several types of cancer [4,5] and also suffer from higher rates of cardiovascular disease [6,7], which has been suggested as a factor in their lower life expectancy [8,9]. There has been less attention to respiratory conditions, such as asthma and COPD. Asthma is a common condition that affects all age groups, having an adverse impact on quality of life and health care costs [10,11]. COPD is a disease condition prevalent in adulthood and is a significant contributor to mortality both in the U.S. and worldwide [12,13]. Asthma and COPD have both been demonstrated to be risk factors for lung cancer in general populations [14,15] and among nonsmokers [16].

At present there is limited information about differences in respiratory disease for Native Hawaiians and Pacific Islanders. A 2010 telephone-based population survey indicated the prevalence of asthma among Native Hawaiian adults was 152 per 1,000 compared with 115 per 1,000 for the state as a whole [17]. Further surveys in 2011 and 2012 showed the prevalence of any lung disease (asthma and/or COPD) was 20.4 per 1,000 for Native Hawaiians compared to 12.7 per 1,000 for the state as a whole [17]. Comparable differences were found in a representative sample of high school students, where asthma prevalence was 38% for Native Hawaiian youth compared with 18% for White adolescents and 22% for Asian-American adolescents [18]. Thus, these ethnic differences may start early on.

Data on emergency department visits in Hawaii [19] and medical records from a California health maintenance organization [20] indicated that Filipinos had an elevated rate of hospitalization for asthma whereas Native Hawaiians had the highest rate of emergency department visits and hospitalizations for COPD and the highest rate of progression from having exclusive COPD to having COPD plus chronic heart failure. However, aside from these studies, information on the prevalence of respiratory disorders among Native Hawaiians and Pacific Islanders remains limited.

Delineating the basis for ethnic differences involves understanding the risk factors for asthma and COPD in the general population. Asthma is a multifactorial condition influenced by biological factors and social and environmental exposures throughout the life course [11] Prominent risk-promoting factors are low socioeconomic status (SES) and poverty. Environmental toxins (e.g., mold, insect pests, air pollution) from housing conditions have a role in producing asthma and risk may also derive from occupational exposures and psychosocial stressors [21,22]. Cigarette smoking and e-cigarette use are also risk factors for asthma [10,18]

and obesity is a risk factor that tends to be independent of these others [23,24]. These interme-diate factors possibly form part of the pathways to health outcomes because disparities in the prevalence of asthma persist even after accounting for SES [11]. For COPD, cigarette smoking is a primary risk factor. However, asthma is a longitudinal risk factor for the development of COPD [25] and, in adulthood, there can be an overlap between the two syndromes [26,27]. Perhaps as a consequence there is an SES gradient for COPD and it shares some risk factors for asthma, including air pollution and occupational exposures [28].

In the present research, we disaggregated participants with Native Hawaiian ancestry from those with non- native Hawaii Pacific Islander ancestry, those with Filipino ancestry, and those with other Asian ancestries for analysis [3]. The data for this research come from two community samples of Hawaii adults, which allows for the characterization of differences in the prevalence of respiratory disease for various ethnic groups and for delineation of behav-ioral factors that are linked to disparities in respiratory disease prevalence. Our primary hypothesis was that Native Hawaiian ethnicity would be related to higher levels of five risk fac-tors—cigarette smoking and e-cigarette use, financial stress, second-hand smoke exposure, and obesity—that can serve as intermediates for the likelihood of adverse health outcomes independent of SES. We expected that Pacific Islanders and Filipinos could share some expo-sures but did not frame specific hypotheses because of a lack of epidemiological data on these groups. After characterizing ethnic differences in respiratory disease prevalence, we performed mediation analyses [29,30] to test for the significance of pathways linking ethnic-group mem-bership to risk-factor exposures and likelihood of respiratory disorder. Results from initial mediation tests with a 2016 survey were tested for replicability with data from a 2018 survey to determine whether observed indirect and direct effects for ethnicity would be replicated across two independent studies.

## Methods

The Behavioral Risk Factor Surveillance Survey (BRFSS) is an annual cross-sectional, random-digit-dialed survey that includes participants aged 18 years and older [31]. The BRFSS uses a multistage sampling design to select a representative sample of the noninstitutionalized adult population in each of the 50 U.S. states. Details on the reliability and validity of the BRFSS sur-vey measures have been summarized elsewhere [32,33]. Since 2011, the methodology has included reaching and interviewing participants on cellular telephones as well as landlines and using a dual-frame survey methodology to improve the reliability and representativeness of the data [34].

We used data from the 2016 and 2018 BRFSS surveys for the state of Hawaii, which has a population of diverse ethnicity. The Hawaii BRFSS has variables for ethnicity that are not in the national survey (e.g., Native Hawaiians disaggregated from all other Pacific Islanders). The survey was conducted with the same basic method and measures each year, though variables could be dropped or added in a given year. (We also analyzed data from the 2017 survey and results were similar but this survey had fewer variables so these results are not reported for space reasons.) The overall response rate for the 2016 Hawaii survey was 43.0%, a figure that compares favorably with other states and national surveys [35], and the response rate for the 2018 survey was 43.6%. The total sample size for the 2016 survey was N = 8,087 cases; for the 2018 survey, it was N = 7,901 cases.

### Measures

**Demographics.**   The survey included items on gender (female/male) and age in years (open- ended). The item on primary ethnicity asked: "Which one of these groups would you

say best represents your ethnicity?" Response options included: Alaska Native/American Indian, Black or African-American, Asian (Chinese, Filipino, Japanese, Other Asian), Native Hawaiian, Pacific Islander, and White. Educational level ("What is the highest grade or year of school you have completed?") had five response options (Grades 1–8, 9–11, 12 years, 1–3 years college, 4 years college or more).

**E-cigarette use.**   A preliminary instruction stated: "Electronic cigarettes and other 'vaping' products include electronic hookahs (e-hookahs), vape pens, e-cigars, and others. These products are battery powered and usually contain nicotine and flavors such as fruit, mint, or candy." An item on ever-use asked, "Have you ever used an electronic cigarette or other electronic 'vaping' product in your entire life," with the response options of Yes, No, or Not Sure. An item on current use asked, "Do you now use electronic cigarettes or other electronic 'vaping' products every day, some days, or not at all?"

**Cigarette smoking.**   A clarifying instruction stated: "For cigarettes do not include electronic cigarettes (e-cigarettes, NJOY, Bluetip), herbal cigarettes, cigars, cigarillos, little cigars, pipes, bidis, kreteks, water pipes (hookahs), or marijuana." The basic item asked, "Have you smoked at least 100 cigarettes in your entire life?" with the response options of Yes, No, or Not Sure. An item on current use asked, "Do you now smoke cigarettes every day, some days, or not at all?"

**Financial stress.**   Financial stress was assessed with two questions: "How often in the past 12 months would you say you were worried or stressed about: Having enough money to pay your rent/mortgage? Having enough money to buy nutritious meals?" Response options were on a 1–5 scale: Never, Rarely, Sometimes, Usually, or Always. The two items were added to form a 10-point scale and the total score was converted for analysis to a 3-point scale (Never/Rarely; Sometimes/Usually; Always).

**Second-hand smoke (SHS) exposure.**   Exposure to second-hand smoke was assessed with two items. The first was: "On how many of the past 7 days did anyone smoke in your home while you were there?" Response options were None, Number of days (open-ended, 1–30 days), or I was not at home in the past 7 days. The second item was: "In the past 7 days, have you been in a car with someone who was smoking?" (Yes/No).

**Body mass index.**   Body mass index (BMI) was assessed with two open-ended items: "About how much do you weigh without shoes?" and "About how tall are you without shoes?" A constructed variable on body mass index with four levels provided by CDC (Underweight, Normal Weight, Overweight, Obese) was used for analysis.

**Respiratory disease.**   Items on respiratory disease followed the stem: "Has a doctor, nurse, or other health professional ever told you that you had any of the following?" Asthma was indexed by the item "(Ever told) you had asthma?" COPD was indexed by "(Ever told) you had chronic obstructive pulmonary disease, C.O.P.D., emphysema or chronic bronchitis?" Responses were Yes/No for each item.

## Analysis methods

Statistical analyses were conducted in September 2022 using SAS 9.4. Data were recoded as necessary so that a higher score meant more of the named quantity (e.g., more smoking, more second-hand smoke exposure). Distributions for demographics, hypothesized mediators, and respiratory outcomes were determined with frequency analyses. For e-cigarette use cigarette smoking, dichotomous items for ever/current use were used to construct 4-level indices for these constructs (never used, used previously but not now; use now some days, use now every day). A continuous measure for age was recoded to 8 deciles. Educational level was recoded to four levels, collapsing grade school/some high school. For secondhand smoke exposure, the

home item was dichotomized (No exposure vs. any exposure) and a 4-level score was constructed (No exposures, 1 exposure—car only, 1 exposure—home only), 2 exposures (car + home).For analysis of ethnicity, two groups (Alaska Native and African American) were dropped because of small sample sizes, and four binary codes were constructed that contrasted Native Hawaiians, Filipinos, Pacific Islanders, and Whites against Asian Americans (Japanese, Chinese, and Other Asians) as the reference group. These three groups were combined because they had uniformly low levels of risk factors, substance use, and respiratory disease relative to other groups. Because other Asians are a heterogeneous group, we performed sensitivity analyses using only Japanese and Chinese as the reference group and found that the results were essentially the same; hence we performed multivariable analyses based on the combined three ethnicities as the reference group BMI was recoded to three levels, collapsing underweight and normal weight. For criterion variables, analyses used binary indices for the ever-diagnosis of asthma and/or COPD Prevalence estimates for the study variables were computed using SAS Proc Surveyfreq including stratification and weighting variables. Cross-tabulation of the ethnicity codes with the respiratory variables was conducted to determine whether there were ethnic differences in disease patterns; the cell chi-square was used to test for significant departure from the expected cell frequency based on the marginal distributions for the ethnic group index and the disease variable. Correlations among the demographic variables were computed using standard procedures.

The mediation hypotheses were tested with a structural modeling analysis using Mplus version 8.6. A formal model of the relations among the variables specified four binary indices for ethnicity as exogenous (i.e., not predicted by any prior variable in the model) together with three covariates (age, sex, and education) that were posited as correlated with ethnicity but not part of the causal process. The correlations among all the seven exogenous variables were estimated as part of the modeling process. Five hypothesized mediator variables (e-cigarette use, cigarette smoking, financial stress, SHS exposure, and BMI) were specified as intermediate between ethnicity and the respiratory outcomes, with paths specified from each of the exogenous variables to all the hypothesized mediators. The five mediators (four in 2018) were specified with covariances of their residual terms. The criterion variables were asthma and COPD. These two variables were specified with a covariance of their residual terms. The fit of the hypothesized model to the data was evaluated with the chi-square statistic, the Comparative Fit Index (CFI), and the Root Mean Square Error of approximation (RMSEA).

The structural model was analyzed with asthma and COPD specified as categorical variables and the mediators specified as ordinal (i.e., ordered categorical) variables. The coefficients for the model were estimated using the WLSMV method (Weighted Least Squares with mean- and variance-adjusted chi-square test statistic). We initially analyzed a saturated model, with all paths from exogenous variables to mediators and all paths from mediators to outcomes. Nonsignificant paths were then dropped from the model and a new model was re-estimated. If appropriate, direct effects (i.e., a path from an exogenous variable to an outcome not involving a mediator) were introduced based on modification indices > 20 (approximately $p < .0001$). The final model was then re-estimated with the significant paths. The significance of indirect (i.e., mediated) effects for ethnicity was evaluated by estimating the Critical Ratio (CR), the cross-product of two paths over its estimated standard error; whose value is analogous to a z-test. Standard errors were computed using the bootstrapped method with 1000 iterations [36].

## Results

A preliminary analysis indicated that cigarette smoking prevalence was comparable at all age levels but e-cigarette use was infrequent among persons 80 years or older. Accordingly, the

analytic samples included only participants between 18 and 79 years of age (analytic n's = 7,477 for 2016 data and 7,414 for 2018 data). Descriptive statistics are in Table 1. Approximately 75% of the participants had not used e-cigarettes, 20% were former users, and 4%-8% were current users (more so in 2018 than in 2016). For cigarette smoking, approximately 60% of the participants were nonsmokers, 25% were former smokers, and 15% were current smokers.

For race/ethnicity, Whites were the largest group and there were also substantial proportions of Native Hawaiians, Filipinos, and Asian-Americans. Characteristics of the samples were generally comparable across years except there were more Native Hawaiians and Pacific Islanders and fewer Whites in the 2018 survey compared to the 2016 survey. Women and men were equally represented in the sample, and their educational level was predominantly above the high school graduate level. Moderate to high financial stress was reported by a substantial proportion of participants (2016 survey only), and approximately 15% of the participants had SHS exposure. The overall prevalence of obesity (approximately 25%) was lower than national rates [37] but there was considerable variation across ethnic groups. Prevalence was approximately 17% for diagnosed asthma and 4% for diagnosed COPD.

## Cross-tabulation and correlation analyses

Cross-tabulation of the ethnicity indices with asthma and COPD indicated ethnic differences as indexed by significant cell chi-squares (Table 2). Asthma was overrepresented among Native Hawaiians (27%) and was underrepresented for the three Asian-American groups (12–15%). Whites tended to be slightly below the marginal proportion while Filipinos and Pacific Islanders were not significantly different. COPD was significantly overrepresented among Native Hawaiians (7%) and underrepresented for the three Asian American groups (2–4%). Whites were sometimes, but not always, below the marginal rate, while Filipinos and Pacific Islanders were close to the marginal rate. Thus, there was a notable ethnic disparity in respiratory disease, with Native Hawaiians elevated compared to other groups in Hawaii.

Cross-tabulation of the asthma and COPD variables indicated a significant relationship, found both for 2016 data, $\chi^2$(1 df) = 364.4, p < .0001, and for 2018 data, $\chi^2$(1 df) = 344.9, p < .0001. For 2016data, 15.0% of participants without COPD had been diagnosed with asthma compared to 52.2% of participants with COPD, and for 2018 data, 14.3% of participants without COPD had been diagnosed with asthma compared to 51.9% of participants with COPD. Thus, having asthma was associated with a 3.5 times greater likelihood of COPD compared to never having asthma. This association was included as part of the structural modeling procedure.

Correlations among the exogenous variables for the structural model (S1 Table) were similar for the 2016 data and 2018 data. Correlations among the variables were generally low. Native Hawaiian and Pacific Islander ethnicity were associated with lower educational level (range r = -.10 and r = -.21, respectively). Whites in the sample tended to be older than Native Hawaiians and Pacific Islanders. These correlations were included as covariates in the structural modeling procedure.

## Structural modeling of mediation in 2016 data

In estimating the structural model for 2016 data, nonsignificant paths were dropped from the initial saturated model except the five hypothesized paths for Native Hawaiian ethnicity, which were forced to remain in the model. Two direct effects for asthma were added based on modification indices: a direct effect with a positive sign from Native Hawaiian to asthma (i.e., other things equal, Native Hawaiians had a greater likelihood of asthma) and a direct effect from

**Table 1. Descriptive statistics for Hawaii BRFSS 2016 and 2018 analytic samples.**

| | 2016 data | 2018 data |
|---|---|---|
| Variable | | |
| Female | 48.9% | 49.4% |
| Age in years | | |
| 18–29 | 21.0% | 20.5% |
| 30–39 | 20.6% | 19.3% |
| 40–49 | 14.9% | 15.7% |
| 50–59 | 17.2% | 17.1% |
| 60–69 | 16.6% | 16.5% |
| 70–79 | 9.8% | 10.9% |
| Race/Ethnicity[a] | | |
| Japanese | 19.1% | 16.9% |
| Chinese | 6.0% | 6.3% |
| Other Asian | 2.7% | 6.8% |
| Native Hawaiian | 13.4% | 19.5% |
| Filipino | 18.6% | 15.7% |
| Pacific Islander | 3.2% | 6.7% |
| White | 37.2% | 28.1% |
| Education | | |
| Grade school | 8.2% | 7.8% |
| HS graduate | 28.6% | 29.0% |
| Some college | 34.5% | 33.6% |
| College graduate | 28.9% | 29.7% |
| Financial stress | | |
| Low | 73.6% | –[b] |
| Moderate | 21.8% | –[b] |
| High | 4.5% | –[b] |
| SHS exposure | | |
| None | 84.9% | 83.1% |
| 1 exposure (car) | 6.4% | 6.6% |
| 1 exposure (home) | 5.1% | 6.1% |
| 2 exposures (car + home) | 3.6% | 4.2% |
| Body mass index | | |
| Normal | 41.2% | 39.3% |
| Overweight | 34.2% | 34.9% |
| Obese | 24.6% | 25.8% |
| E-cigarette use | | |
| Nonuser | 76.6% | 70.9% |
| Former user | 18.9% | 21.5% |
| Current user[c] | 4.5% | 7.6% |
| Cigarette smoking | | |
| Nonsmoker | 61.7% | 60.4% |
| Former smoker | 24.7% | 25.5% |
| Current smoker[c] | 13.6% | 14.0% |
| Respiratory disease | | |
| Ever Dx asthma | 17.7% | 16.6% |

(*Continued*)

Table 1. (Continued)

|  | 2016 data | 2018 data |
|---|---|---|
| Ever Dx COPD | 3.7% | 3.6% |

*Note*: N for analytic samples is 7,477 for 2016 data, 7,414 for 2018 data. Frequencies are for weighted analyses.

[A] Blacks and Alaska Natives are excluded from tabulation because of small cell sizes.

[B] Variable was not included in this survey.

[C] Includes use 'some days' and 'every day".

male sex with a negative sign (i.e., males had less likelihood of asthma). Three direct effects for COPD were added based on modification indices: a direct effect with positive sign from Native Hawaiian to COPD, a direct effect for Whites (same), and a direct effect for age with a positive sign (i.e., other things equal, older persons had a greater likelihood of COPD). The final model had $\lambda^2$ (21 df, N = 7,058) of 46.64, CFI of 0.99, and RMSEA of 0.013 (CI .008-.018), all indices representing a good fit of the model to the data.

The structural model is presented graphically ([Fig 1]) with standardized coefficients, all p < .0001 except for the path from Native Hawaiian to e-cigarette use, the direct effects for Native Hawaiian and White ethnicity, and the path from Pacific Islander ethnicity to secondhand smoke exposure. Residual correlations among the mediator constructs in the model (included

Table 2. Cross-tabulation for Asthma and COPD (%) by race/ethnicity, for BRFSS 2016 and 2018.

| Asthma |  |  | 2016 Data |  |  | 2018 data |
|---|---|---|---|---|---|---|
|  | No | Yes | Marginal N/[%] | No | Yes | Marginal N/[%] |
| Group |  |  |  |  |  |  |
| Japanese | 85% | 15%* | 1,308 [18%] | 85% | 15% | 916 [13%] |
| Chinese | 87% | 13%+ | 340 [5%] | 84% | 16% | 306 [4%] |
| Other Asian | 88% | 12%+ | 149 [2%] | 84% | 16% | 346 [5%] |
| White | 85% | 15%* | 3,130 [44%] | 87% | 13%*** | 2,616 [38%] |
| Filipino | 83% | 17% | 968 [14%] | 85% | 15% | 842 [12%] |
| Pac. Islander | 87% | 13% | 184 [3%] | 85% | 15% | 404 [6%] |
| Nat. Hawaiian | 73% | 27%*** | 1,043 [15%] | 75% | 25%*** | 1,400 [20%] |
| Marginal % | 83% | 17% |  | 84% | 16% |  |
| Total N | 5,916 | 1,206 | 7,122 [100%] | 5,720 | 1,110 | 6,830 [100%] |
| COPD |  |  |  |  |  |  |
| Japanese | 96% | 4%** | 1,312 [18%] | 97% | 3%** | 910 [13%] |
| Chinese | 98% | 2%** | 342 [5%] | 96% | 4% | 305 [4%] |
| Other Asian | 98% | 2%+ | 150 [2%] | 97% | 3% | 345 [5%] |
| White | 94% | 6%* | 3,121 [44%] | 95% | 5% | 2,612 [38%] |
| Filipino | 96% | 4% | 968 [14%] | 97% | 3% | 841 [12%] |
| Pac. Islander | 96% | 4% | 182 [3%] | 95% | 5% | 400 [6%] |
| Nat. Hawaiian | 93% | 7%*** | 1,042 [15%] | 94% | 6%** | 1,396 [20%] |
| Marginal % | 95% | 5% |  | 95% | 5% |  |
| Total N | 6,729 | 388 | 7,117 [100%] | 6,488 | 321 | 6,809[100%] |

*Note*: Distributions based on analytic samples. Asterisks indicate cell departure from expectation: + p < .10

* p < .05

** p < .01

*** p < .001.

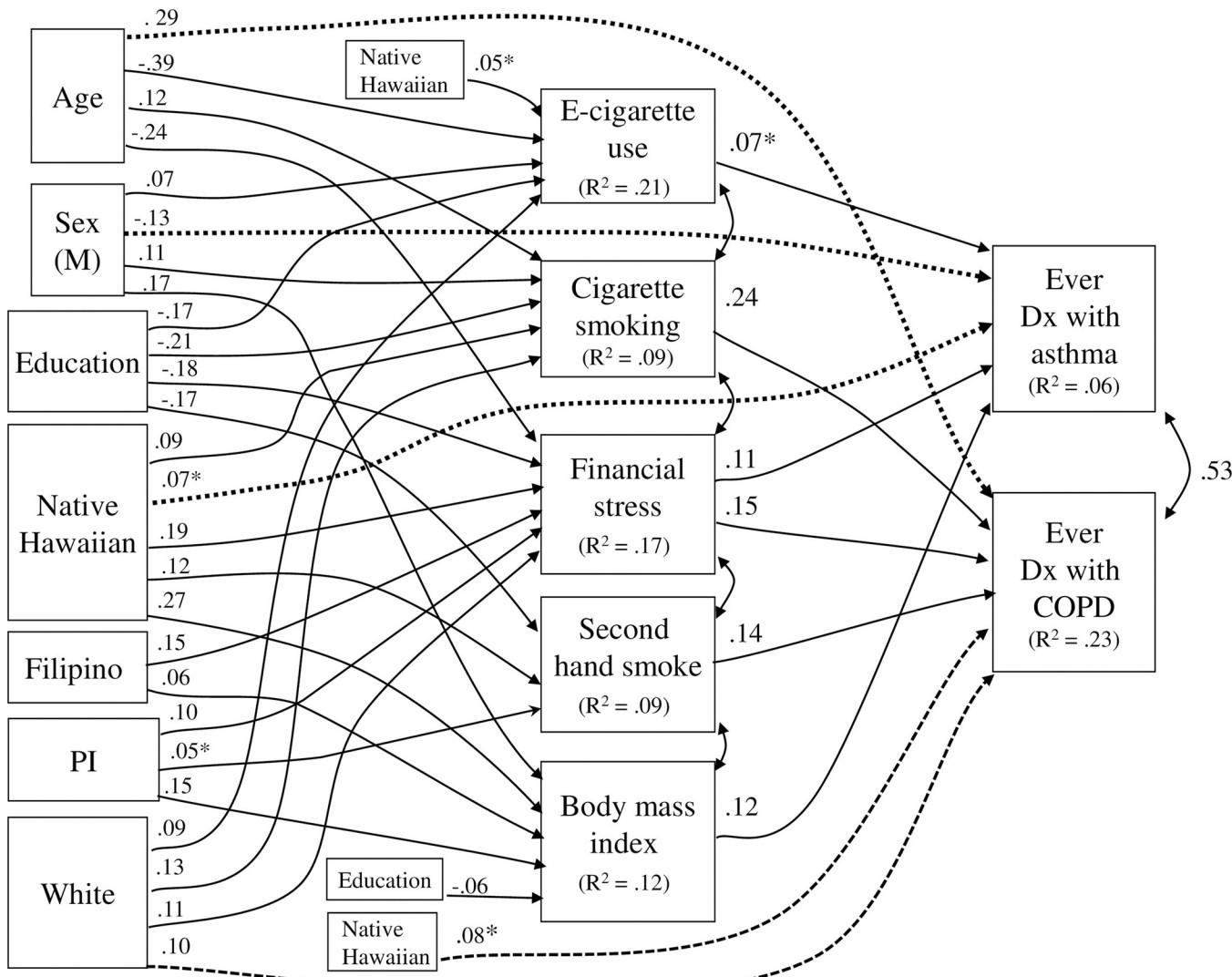

**Fig 1. Structural model for ethnicity and respiratory disorder, for 2016 Hawaii BRFSS data.** Analytic N = 7,078. Nat Hawn = Native Hawaiian; Dx = diagnosis. For ethnicity indicators, reference group is Asian-American. Values are standardized coefficients. Coefficients are significant at p < .0001 unless otherwise noted; * indicates p < .01. Straight single-headed arrows represent paths; curved double-headed arrows represent covariances. Dashed arrows indicate direct effects. Included in the model but excluded from the figure for graphical simplicity are correlations among the exogenous variables (see S1 Table) and correlations among the residuals for the five hypothesized mediator variables (see S2 Table), and paths from age to secondhand smoke exposure, β = -.16, and from age to body mass index, β = .0, p < .01.. $R^2$ values in rectangles indicate the variance accounted for in a given construct by all variables to the left of it in the model. Excluded from the figure for graphical simplicity are paths (all p < .01) to SHS exposure from age (beta = -.07) and Pacific Islander ethnicity (beta = .05) and to body mass index from age (beta = .05) and education (beta = -.06).

in the model but not included in the figure) are in S2 Table. In these correlations, e-cigarette use was positively correlated with cigarette smoking and with SHS exposure. Other residual correlations were generally low.

**Paths from Native Hawaiian ethnicity to mediators.** There were significant paths from Native Hawaiian ethnicity to a greater likelihood of e-cigarette use, cigarette smoking, financial stress, second-hand smoke exposure, and higher body mass index (controlling for educational level). Thus, the hypothesis that Native Hawaiian ethnicity would be related to the identified risk factors was supported. There were also direct effects (not involving any of the mediator

variables) from Native Hawaiian ethnicity to a greater likelihood of asthma (p < .001) and a greater likelihood of COPD (p < .001).

**Paths for other ethnic groups.** There were significant paths from Filipino ethnicity to greater financial stress and higher body mass index, and similar paths for Pacific Islander ethnicity. There was also a path from Pacific Islander ethnicity to more secondhand smoke exposure. Whites (relative to Asian-Americans) were more likely to engage in e-cigarette use and cigarette smoking, to have financial stress, and (independently) to have a higher likelihood of COPD.

**Paths for demographic variables.** Older age was associated with less e-cigarette use and more cigarette smoking and second hand smoke exposure. Older age was also related to less financial stress and had a direct effect to greater likelihood of having COPD. Male sex was associated with more substance use and higher BMI and had a direct effect to lower likelihood of having asthma. Educational level was associated with lower levels of all five risk factors.

**Paths for mediators to outcomes.** Significant paths to asthma were found for e-cigarette use, financial stress, and body mass index, all with positive sign (i.e., contributing to a greater likelihood of diagnosed asthma). Significant paths to COPD with positive sign were found for cigarette smoking, financial stress, and SHS exposure. These paths are independent of the direct effects previously noted for Native Hawaiian ethnicity and other demographic variables. Together the variables in the model accounted for 6% of the variance in asthma and 23% of the variance in COPD.

## Structural modeling of mediation in 2018 data

The structural model for 2018 data (Fig 2) had $\lambda^2$ (18, N = 6,388) of 41.71, CFI of 0.99, and RMSEA of 0.014 (CI .009-.020). Again, all these indices represented a good fit of the model to the data. Almost all the paths found for the 2016 data were replicated in the independent analysis (p < .0001 unless otherwise noted).

## Paths from Native Hawaiian ethnicity to mediators

The previous paths from Native Hawaiian ethnicity to e-cigarette use, cigarette smoking, SHS exposure, and BMI were all replicated. One direct effect, from Native Hawaiian to a greater likelihood of asthma, was also replicated, and an inverse direct effect from sex to COPD (males lower) was now included in the model.

**Paths for other ethnic groups.** Results for other ethnic groups were not as replicable in 2018 data but a previous path from Pacific Islander ethnicity to higher BMI was replicated and paths for Filipinos and Pacific Islanders to SHS exposure were noted. A path for Whites to cigarette smoking was replicated, as was a direct effect to COPD.

**Paths for demographic variables.** Older age was associated with less e-cigarette use and more cigarette smoking, Older age was also related to less SHS smoke exposure and had a direct effect to greater likelihood of having COPD. Male sex was associated with more substance use and higher BMI and had a direct effect to lower likelihood of having asthma. Educational level was associated with lower levels of all the five risk factors. All these results are replications of findings from the analysis of the 2016 data.

**Paths from mediators to outcomes.** Paths from the risk factors to asthma were all replicated, as were paths from cigarette smoking and SHS exposure to COPD; also, new paths from e-cigarettes and BMI to COPD were now included. The new paths for e-cigarettes may be a consequence of the higher prevalence for e-cigarette use in this year. Together the variables in the model accounted for 6% of the variance in asthma and 23% of the variance in COPD, similar to findings for the 2016 data.

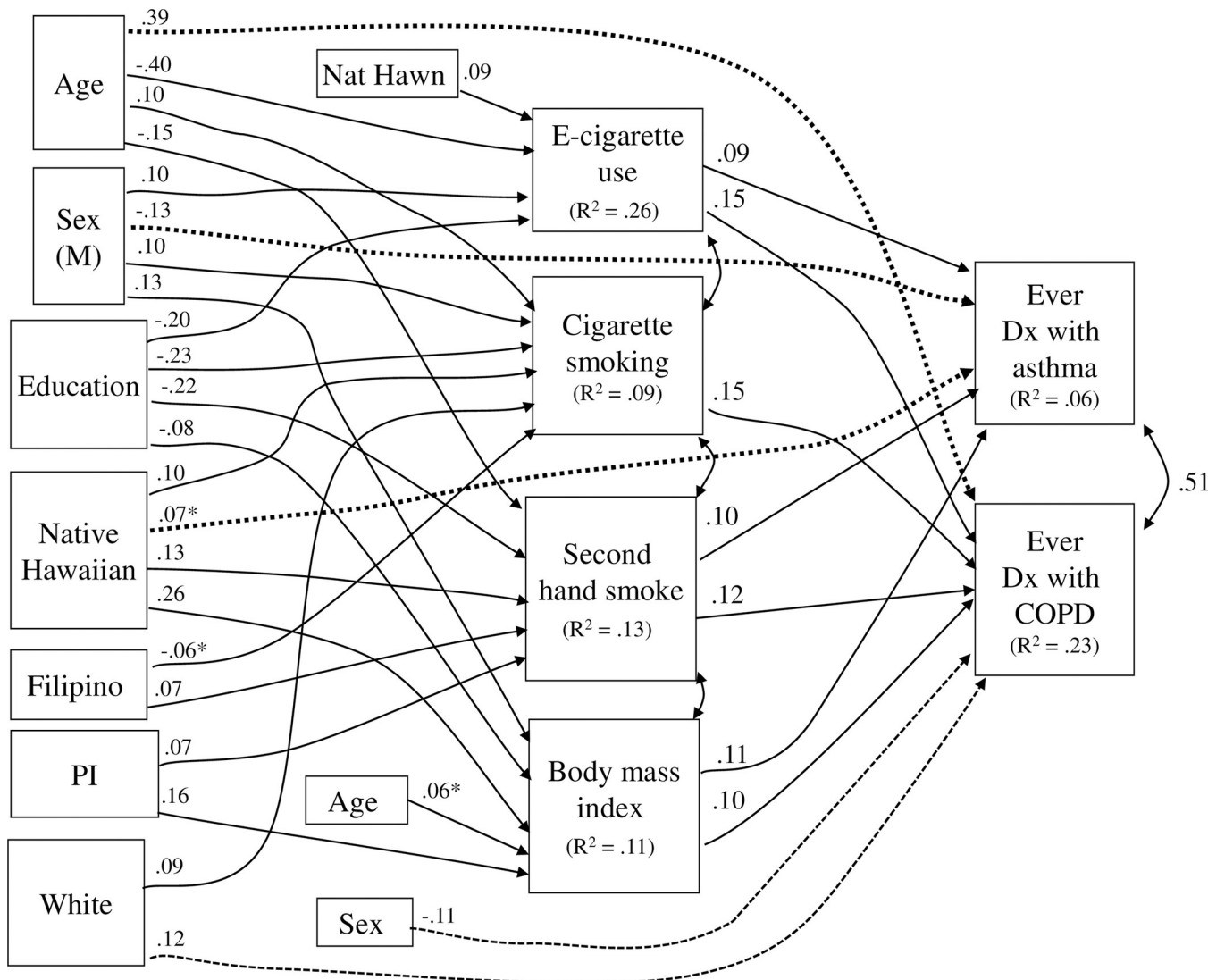

**Fig 2. Structural model for ethnicity and respiratory disorder, for 2018 Hawaii BRFSS data.** Analytic N = 6,388. Nat Hawn = Native Hawaiian; Dx = diagnosis. For ethnicity indicators, reference group is Asian-American. Values are standardized coefficients. Dashed arrows indicate direct effects. For other notation, see Fig 1 caption. Excluded from the figure for graphical simplicity are paths (all p < .01) to body mass index from age (β = .06) and from sex to COPD (β = -.11).

### Significance of indirect effects for ethnicity

**2016 data.** Tests of the indirect effects for Native Hawaiian and other ethnicity variables for 2016 data are presented in Table 3 (left-hand column). For Native Hawaiians, indirect effects to asthma through financial stress and BMI were significant (marginal for e-cigarette use). Indirect effects to COPD through cigarette smoking, financial stress, SHS exposure, and BMI were significant. In addition, direct effects from Native Hawaiian to asthma and COPD were significant. For Filipinos, indirect effects to asthma through financial stress and BMI were significant; in addition, there was a significant indirect effect to COPD through financial stress. For Pacific Islanders, indirect paths to asthma through financial stress and BMI were significant as well as indirect effects to COPD through financial stress and SHS exposure. In this year, there were significant indirect effects for Whites to asthma through e-cigarette use

and financial stress, significant indirect effects to COPD through cigarette smoking and financial stress, and a direct effect to COPD was significant.

**2018 data.** Indirect effects for ethnicity variables (Table 3, right-hand column) were mostly replicated. Six significant indirect effects for Native Hawaiians replicated effects from the 2016 model, and an indirect effect for Native Hawaiians to COPD through BMI was significant. A previous indirect effect for Pacific Islanders (asthma through BMI) was replicated and indirect effects for Pacific Islanders to asthma and COPD through SHS exposure were now significant. A previous indirect effect to COPD for Whites through cigarette smoking was replicated, as was the direct effect to COPD for Whites.

## Indirect effects for education

**2016 data.** Several protective indirect effects were noted for education (i.e., higher education indirectly related to a lower likelihood of respiratory disease through the hypothesized

**Table 3. Critical Ratio and probability (p) levels for tests of indirect and direct effects to respiratory disease for Native Hawaiian and Pacific Islander ethnicity, for 2016 data and 2018 data.**

| Indirect or direct effect | 2016 data | | 2018 data | |
|---|---|---|---|---|
| | Critical ratio | P | Critical ratio | P |
| Nat. Hawaiian = > E-cigarette use = > Asthma | 1.77 | .08 | 2.36 | .02 |
| Nat. Hawaiian = > Financial stress = > Asthma | 3.52 | < .001 | –[a] | – |
| Nat. Hawaiian = > SHS exposure = > Asthma | – | – | 2.68 | .01 |
| Nat. Hawaiian = > BMI = > Asthma | 5.24 | < .0001 | 4.30 | < .0001 |
| Nat. Hawaiian = > Asthma [direct effect] | 3.53 | < .001 | 3.02 | < .01 |
| Nat. Hawaiian = > E-cigarette use = > COPD | – | – | 2.22 | .03 |
| Nat. Hawaiian = > Cig. smoking = > COPD | 4.24 | < .0001 | 2.77 | < .01 |
| Nat. Hawaiian = > Financial stress = > COPD | 3.52 | < .001 | –[a] | – |
| Nat. Hawaiian = > SHS exposure = > COPD | 2.61 | .< .01 | 2.12 | .03 |
| Nat. Hawaiian = > BMI = > COPD | – | – | 3.15 | .001 |
| Nat. Hawaiian = > COPD [direct] | 2.90 | < .01 | – | – |
| Filipino = > Financial stress = >Asthma | 3.40 | < .001 | –[a] | – |
| Filipino = > BMI = >Asthma | 3.16 | .002 | – | – |
| Filipino = > SHS exposure = > Asthma | – | – | 2.21 | .03 |
| Filipino = > Cig. smoking = > COPD | – | – | -2.28 | .02 |
| Filipino = > Financial stress = > COPD | 3.23 | .001 | –[a] | – |
| Filipino = > SHS exposure = > COPD | – | – | 1.83 | .07 |
| Pac. Islander = > Financial stress = > Asthma | 3.40 | < .001 | –[a] | – |
| Pac. Islander = >BMI = > Asthma | 4.85 | < .0001 | 4.22 | < .0001 |
| Pac. Islander = > SHS exposure = > Asthma | – | – | 2.35 | .02 |
| Pac. Islander = > Financial stress = > COPD | 3.21 | .001 | –[a] | – |
| Pac. Islander = > SHS exposure = > COPD | 2.02 | .05 | 1.89 | .06 |
| Pac. Islander = > BMI = > COPD | – | – | 3.03 | .002 |
| White = > E-cigarette use = > Asthma | 2.37 | .01 | – | – |
| White = > Financial stress = > Asthma | 2.87 | < .01 | – | – |
| White = > Cig. Smoking = > COPD | 5.19 | < .0001 | 2.65 | .01 |
| White = > Financial stress = > COPD | 2.80 | < .01 | –[a] | – |
| White = > COPD [direct] | 3.08 | .002 | 2.94 | < .01 |

*Note.* The Critical Ratio is analogous to a *z* test. Indirect effect estimates are based on bootstrap standard errors.

Simple dashes (——) indicate there was no path from the variable to the mediator in this year, hence no possibility of an indirect effect. Dashes with superscript ([A]—) indicate variable was not included in the survey in this year.

mediator variables). The results of significance tests for 2016 data are in S3 Table. Significant indirect effects of education for asthma were noted through less e-cigarette use, less financial stress, and lower BMI and for COPD through less cigarette smoking and second-hand smoke exposure. Notably, there were no direct effects from education to respiratory disease in this model.

**2018 data.**   Significant indirect effects for asthma (S3 Table) were noted through less e-cigarette and lower BMI, replicating findings from the 2016 data. A significant indirect effect through less SHS exposure among more educated persons was also noted. Replicating findings from the 2016 study, significant indirect effects for COPD were noted through less cigarette smoking and second-hand smoke exposure, and significant indirect effects were also noted through less e-cigarette use and lower BMI. Again, there were no direct effects from education to respiratory disease outcomes.

## Discussion

This research was conducted to gain more understanding of the basis for disparities in respiratory disorders for Native Hawaiians and Pacific Islanders. We analyzed data from a statewide representative sample of Hawaii adults using measures with known reliability and validity. Our initial analyses indicated a significant differential in respiratory disease for Native Hawaiians, who have elevated rates for both asthma and COPD compared to members of other ethnic groups in Hawaii. The mediation analyses specified a formal model of possible pathways from ethnic group membership to risk factors for respiratory disease and evaluated the significance of these pathways for both asthma and COPD. Our basic prediction, that the disease elevation for Native Hawaiians would be linked to greater exposure to the five hypothesized risk factors, was confirmed in analyses for the 2016 survey and these findings were largely replicated in analyses of the independent 2018 survey. The data also indicated a significant association between having had asthma and having COPD, consistent with data from other studies [26,27].

The results supported our hypothesis that Native Hawaiian ethnicity would be related to greater exposure to several risk factors for respiratory disease. The mediation analysis confirmed that there were independent paths from Native Hawaiian ethnicity to a higher frequency of e-cigarette use and cigarette smoking, as well as to individual physiological variables (body mass index) and social-environmental variables (financial stress and SHS exposure). The disparities in respiratory disease were linked not just to one variable but to a range of variables from different domains. This supports a social-ecological model by pointing to origins in social as well as individual risk factors [11,28,38]. It should be noted that SES, indexed through educational level, had no direct effects to respiratory outcomes in the structural models, indicating that associations of SES with disease outcomes were mediated in these samples through the risk factors we specified. While findings for ethnic-group membership were independent of education, supplemental analyses (S3 Table) clarified how education itself operates as a health-protective factor in this population, through pathways to less stress and substance use and to more favorable physical status.

Our mediation analyses established that the hypothesized mediator variables served, in part, to contribute to the observed ethnic disparities for respiratory disease, particularly so for Native Hawaiians but also for Filipinos and Pacific Islanders. The results are consistent with previous basic research on predictors of asthma and COPD [11,28] but extend this research by showing specific pathways from ethnic group membership to specified disease predictors, knowledge that can provide a basis for more informed design of prevention and treatment programs. The results also extend previous research by showing that an emerging phenomenon, electronic cigarettes, constitute an independent risk factor for respiratory disease [24,39].

It should be noted that the results included protective effects, for example for education. This provides a public health perspective for disease prevention, suggesting that respiratory disease rates can be reduced through higher educational attainment and/or improved access to quality educational opportunities. Native Hawaiians and Pacific Islanders often have lower access to educational opportunities, especially post-secondary education. The present findings point to a need for studying resilience and protective effects for Native Hawaiian ethnicity, studying Native Hawaiian values and cultural themes such as supportive family or community relationships [40]. These have been demonstrated in other research to be important resilience factors for mental health and substance use among Native Hawaiians and other racial/ethnic groups [41–43]. Supportive relationships may also be important as buffers against racial/ethnic discrimination [44,45] and adverse economic or neighborhood conditions [46–48].

The structural modeling analyses indicated that the variables in the model accounted for 6% of the variance in asthma and 23% of the variance in COPD. The difference in explained variance probably arises because asthma is a multifactorial disease, with contributions from genetic makeup, housing conditions, environmental factors, and adulthood occupational exposures [11,21,22]. For COPD, in contrast, the major primary predictors are age and history of cigarette smoking [28], consistent with the present results. However, the analyses showed that other factors, including SHS exposure and obesity, also contribute to the likelihood of COPD in this population. Psychosocial stress arising from adverse childhood experiences, exposure to community violence, or difficult neighborhood conditions is also associated with an increased risk for asthma or asthma exacerbations [49–53]. The present findings illustrate the importance of multimodal measures tapping a range of domains that may contribute to a disease outcome.

A conceptual issue with the present research is that while it demonstrates disparities in exposure to risk factors for respiratory disease, it does not prove how the differential exposures came about in the first place. For example, differences in cigarette smoking can have roots in historical trauma [54–56], or present-day experiences of discrimination and governmental policies that restrict access to higher education opportunities or to secure employment and safe housing [44,51,57]. Understanding of health disparities requires research designed to illuminate the origins of social conditions that contribute to differences in health status today [58].

## Limitations

Although the present research was based on large representative samples and showed replicated findings, some aspects present possible limitations. The measures were relatively simple ones, and the data on asthma and COPD were based on self-report, hence some limitations in accuracy are possible. The analyses controlled for socioeconomic status by including education in the models but other indices of SES, such as occupational status, may also be significant for health disparities [28,59]. The data were cross-sectional and do not definitively demonstrate temporal relationships. This might pose an issue for the financial stress measures, which could be consequences of illness (because of unemployment). However, a considerable body of research has linked psychosocial stressors with respiratory disease [11,49–53,60] hence reverse causation cannot be assumed out of hand. Finally, although the measures for asthma and COPD were based on diagnosis by health professionals, the present studies did not include biological measures (e.g., lung function indices), and further social-ecological studies using direct examination or biological measures would be desirable [28,54,61,62].

## Conclusions

Two studies based on random samples of the Hawaii population established that Native Hawaiians have elevated rates of asthma and COPD. We found that Native Hawaiian ethnicity

is related to higher exposure to five risk factors for respiratory disease, and to some extent the same is true for Pacific Islanders and Filipinos. The elevation in disease rates for Native Hawaiians is partly accounted for by greater risk-factor exposure across multiple domains of variables, controlling for socioeconomic status. The results support a social-ecological model through showing disease vulnerability linked to social as well as individual factors, and possibly to historical ones as well. Programs are needed to reduce exposure of Native Hawaiians and Pacific Islanders to risk factors. Further research is needed to illuminate how resilience factors in the Native Hawaiian community, including strong family support and community networks, operate to buffer the impact of current life stressors.

## Supporting information

**S1 Table. Correlations of exogenous variables, for 2016 and 2018 Hawaii BRFSS data.**
(DOCX)

**S2 Table. Residual correlations of mediator variables in structural models, for 2016 data and 2018 data.**
(DOCX)

**S3 Table. Indirect effects (Critical Ratio and p) for education, for 2016 and 2018 Hawaii BRFSS data.**
(DOCX)

## Acknowledgments

We thank the survey participants for their contribution and the staff of the Hawaii State Department of Health for facilitating access to the data.

## Author Contributions

**Conceptualization:** Thomas A. Wills, Joseph Keawe'aimoku Kaholokula.

**Data curation:** Ian Pagano.

**Formal analysis:** Thomas A. Wills, Joseph Keawe'aimoku Kaholokula, Pallav Pokhrel, Ian Pagano.

**Methodology:** Pallav Pokhrel, Ian Pagano.

**Writing – original draft:** Thomas A. Wills.

**Writing – review & editing:** Thomas A. Wills, Joseph Keawe'aimoku Kaholokula, Pallav Pokhrel, Ian Pagano.

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
