## [Decision Letter · Decision Letter 0]

14 Jul 2023

PONE-D-23-09224Ethnic differences in respiratory disease for Native Hawaiians and Pacific Islanders: Analysis of mediation processes in two community samplesPLOS ONE

Dear Dr. Wills,

Thank you for submitting your manuscript to PLOS ONE. After careful consideration, we feel that it has merit but does not fully meet PLOS ONE’s publication criteria as it currently stands. Therefore, we invite you to submit a revised version of the manuscript that addresses the points raised during the review process.

We look forward to receiving your revised manuscript.

Kind regards,

Yasunori Sato

Academic Editor

PLOS ONE

Journal Requirements:

Reviewers' comments:

Reviewer's Responses to Questions

**Comments to the Author**

1. Is the manuscript technically sound, and do the data support the conclusions?

Reviewer #1: Yes

Reviewer #2: Yes

2. Has the statistical analysis been performed appropriately and rigorously? 

Reviewer #1: Yes

Reviewer #2: Yes

3. Have the authors made all data underlying the findings in their manuscript fully available?

Reviewer #1: Yes

Reviewer #2: Yes

4. Is the manuscript presented in an intelligible fashion and written in standard English?

Reviewer #1: Yes

Reviewer #2: Yes

5. Review Comments to the Author

Reviewer #1: There are minor comments for clarifications by the authors on one of the tables.

Table 1 Page 16 - Merge the cell with empty one above it.

Table 1 Pg 17 - What is this value (4.4) meant to be for? Is it an alignment problem?

Reviewer #2: manuscript is written as per guidelines statistics are well explained the title and aims and objectives of the study are met.The study is primarily of respiratory diseases in the overlooked ethnic groups.

6. PLOS authors have the option to publish the peer review history of their article (what does this mean?). If published, this will include your full peer review and any attached files.

Reviewer #1: **Yes: **Prof. Tanimola Makanjuola Akande

Reviewer #2: **Yes: **Dr Sumer S Choudhary

---

## [Author Response · Author response to Decision Letter 0]

14 Aug 2023

We thank the editor and reviewers for their feedback on the Ms. We respond to the editorial comments as follows: 

1 Style requirements: We have checked the journal’s style requirements and believe it is concordant with all of them. 

2. Ethics statement: We include the full name of the IRB that approved this research: the Committee on Human Studies (=IRB) of the University of Hawaii at Manoa. We note that our research was designated as exempt. The original study (the Behavioral Risk Factor Surveillance Survey or BRFSS) obtained informed consent from participants but we used deidentified data provided by the Hawaii State Department of Health and the authors did not have access to any data that could identify individual participants.

3. Availability of data: We used state-level data for Hawaii containing detailed information on race/ethnicity; the data were obtained from the Hawaii Health Data Warehouse (who got it from the Centers for Disease Control). The Hawaii Health Data Warehouse (hereafter, HHDW) will provide BRFSS data to anyone who asks for it; you just have to provide a list of the variables you need and the names of the person(s) who will work with the data. That is what we meant when we said in the previous submission that the data are available on request. 

There are some restrictions on the data provided: HHDW will not release data on zip codes or certain other demographic information that could be used to identify an individual participant. Zip codes are restricted because Hawaii is a relatively small state with diverse ethnicity, and a person who had data on ethnicity and zip code could possibly identify a participant through deductive identification (e.g., the only female Tongan in the Kona Zip code). If there are any questions about this they can be directed to Dr Tonya Lowery St. John (tlowerystjohn@hhdw.org).

This is a case where the data are owned by a third party: the Hawaii State Department of Health, Chronic Disease Prevention and Health Promotion Division. Because of the third-party ownership, individual users may only use the data for their own research or public health purposes. However, we note that the State Department of Health supports the HHDW, which stores and organizes data from several Hawaii health surveys (including the BRFSS) and makes the data available to users for analysis. In addition to providing record-level data, HHDW also provides original surveys, data codebooks, and CDC technical reports that describe the research methods in detail. Thus it is essentially a public repository.

Because of the third-party ownership issues, we are not posting the data with the Ms submission. 

However, we note that the same BRFSS data we used are available from HHDW, hence other researchers would be able to access and analyze the data sets in the same manner as we did. Accordingly we have identified the HHDW in the Ms as the source of the data and have cited CDC publications that describe the research methods in detail. This would enable anyone who wanted to replicate the study findings to do so.

4. References: There are no changes to the reference list. As far as we know, none of the cited papers have been retracted. 

Now, we respond to the reviewer comment as follows: 

1. Question about entries for secondhand smoke (SHS) exposure in Table 1: There was a glitch here. We checked the data layout and discovered that one item in the 2016 data had not been requested, when it was in fact in the survey. Accordingly we requested and obtained the missing item and rescored the 2016 variable for secondhand smoke exposure so it used the same procedure as was employed for the 2018 data. We then reran the descriptive statistics and the structural modeling and mediation analyses for the 2016 data. Table 1 now contains complete data for four levels of SHS exposure for both years. There are now four completed cells for this variable for each assessment and it is evident that mean levels of exposure were comparable across years. In addition, it is now clear that the value questioned by the reviewer is the mean score for the highest level of exposure in the 2018 data, corresponding to the mean score for the highest level of exposure in the 2016 data. 

The rescoring of one variables resulted in some minor changes to the results, as some paths for SHS exposure were now stronger in magnitude or were now significant when they had previously been marginal. We have revised one part of the Method section to describe the parallel scoring procedures for SHS exposure and have noted in the Results section (text and Table 3) any findings that were different from the previous version. Our conclusions are not greatly different from the previous version but we were able to make some stronger statements about the adverse effects of SHS exposure.

---

## [Editor Report · Decision Letter 1]

16 Aug 2023

Ethnic differences in respiratory disease for Native Hawaiians and Pacific Islanders: Analysis of mediation processes in two community samples

PONE-D-23-09224R1

Dear Dr. Wills,

We’re pleased to inform you that your manuscript has been judged scientifically suitable for publication and will be formally accepted for publication once it meets all outstanding technical requirements.

Kind regards,

Yasunori Sato

Academic Editor

PLOS ONE

---

## [Editor Report · Acceptance letter]

18 Aug 2023

PONE-D-23-09224R1 

Ethnic differences in respiratory disease for Native Hawaiians and Pacific Islanders: Analysis of mediation processes in two community samples 

Dear Dr. Wills:

I'm pleased to inform you that your manuscript has been deemed suitable for publication in PLOS ONE. Congratulations! Your manuscript is now with our production department. 

Kind regards, 

on behalf of

Dr. Yasunori Sato 

Academic Editor

PLOS ONE